

# Implementation of a machine-learned gas optics parameterization in the ECMWF Integrated Forecasting System: RRTMGP-NN 2.0

Peter Ukkonen[1] and Robin J. Hogan[2,3]

[1]Danish Meteorological Institute
[2]European Centre for Medium-Range Weather Forecasts, Reading, UK
[3]Department of Meteorology, University of Reading, Reading, UK

**Correspondence:** Peter Ukkonen (puk@dmi.dk)

**Abstract.** Radiation schemes are physically important but computationally expensive components of weather and climate models. This has spurred efforts to replace them with a cheap emulator based on neural networks (NN), obtaining large speed-ups, but at the expense of accuracy, energy conservation and generalization. An alternative approach which is slower but more robust than full emulation is to use NNs to predict optical properties, but keep the radiative transfer equations. Recently, NNs

were developed to replace the RRTMGP gas optics scheme, and shown to be accurate while improving speed. However, the evaluations were based solely on offline radiation computations.

In this paper, we describe the implementation and prognostic evaluation of RRTMGP-NN in the Integrated Forecasting System (IFS) of the European Centre for Medium-Range Weather Forecasts (ECMWF). The new gas optics scheme was incorporated into ecRad, the modular ECMWF radiation scheme. Using a hybrid loss function designed to reduce radiative

forcing errors, and an early stopping method based on monitoring fluxes and heating rates with respect to a line-by-line benchmark, new NN models were trained on RRTMGP $k$-distributions with reduced spectral resolutions. Offline evaluation of the new NN gas optics, RRTMGP-NN 2.0, shows a very high level of accuracy for clear-sky fluxes and heating rates. For instance, the RMSE in shortwave surface downwelling flux is 0.78 W m$^{-2}$ for RRTMGP and 0.80 W m$^{-2}$ for RRTMGP-NN in a present-day scenario, while upwelling flux errors are actually smaller for the NN. Because our approach does not affect

the treatment of clouds, no additional errors will be introduced for cloudy profiles. RRTMGP-NN closely reproduces radiative forcings for 5 important greenhouse gases across a wide range of concentrations such as 8x CO2.

To assess the impact of different gas optics schemes in the IFS, four 1-year coupled ocean-atmosphere simulations were performed for each configuration. The results show that RRTMGP-NN and RRTMGP produce very similar model climates, with the differences being smaller than those between existing schemes, and statistically insignificant for zonal means of

single-level quantities such as surface temperature. The use of RRTMGP-NN speeds up ecRad by a factor of 1.5 compared to RRTMGP (the gas optics being almost 3 times faster), and is also faster than the older and less accurate RRTMG which is used in the current operational cycle of the IFS.



## 1    Introduction

Although atmospheric radiation is well understood and very accurate solutions are available, atmospheric models need to settle
for a trade-off between the accuracy and cost of radiation computations. This trade-off can be controlled via many factors like
the temporal and spatial frequency of computations (Hogan and Bozzo, 2018), simplifying assumptions (e.g. neglecting 3D
effects), and spectral resolution (Hogan and Matricardi, 2020). To reduce the latter, most modern radiation schemes use the
correlated-$k$-distribution method (e.g., Goody et al., 1989) since it allows broadband fluxes to be computed with high accuracy
using only $O(10^2 - 10^3)$ quadrature points, compared with $O(10^6 - 10^7)$ for line-by-line methods which resolve individual
spectral lines in the absorption spectra of atmospheric gases.

Despite this, computations remain expensive enough that many other of the aforementioned approximations need to be made,
and still coarse-resolution climate simulations can spend half of the total model runtime on radiation (Cotronei and Slawig,
2020). To make better use of computer resources in an era where computer hardware is becoming more heterogenous, and the
gap between hardware peak performance and the performance achived with typical physics codes is probably increasing even
further, machine learning (ML) is a promising way to simultaneously address computational challenges and potentially reduce
model uncertainty by representing sub-grid processes more realistically.

Indeed, interest in the use of ML for parameterization of sub-grid processes has been growing, with a particular focus on
learning convection or unified physics parameterizations from high-resolution simulations (Rasp et al., 2018; Brenowitz and
Bretherton, 2018; Gentine et al., 2018; Brenowitz et al., 2020; Yuval et al., 2021). Using ML specifically for atmospheric
radiation has a long history (Chevallier et al., 1998; Krasnopolsky et al., 2008, 2010; Pal et al., 2019; Liu et al., 2020; Roh
and Song, 2020; Song and Roh, 2021). These studies trained feed-forward neural networks (FNNs) to emulate a physical
radiation scheme, achieving impressive speed-ups typically between 10-100x (although, in many cases relative to outdated
or unoptimized codes). Assessing the level of accuracy achieved is difficult in the absence of common datasets, metrics, and
comparison against benchmark computations, but reported errors are typically too high to be considered for use in the ECMWF
weather forecast model. For instance, shortwave fluxes in Song and Roh (Figure 1, 2021) had root-mean-square-errors (RMSE)
of roughly 15 W m$^{-2}$ in offline evaluation against the original scheme. To our knowledge, the ability of FNN-based radiation
emulators to compute the radiative forcing due to changes in greenhouse gases has not been evaluated, so such schemes must
be considered unsuitable also for climate models until they are shown to do so accurately.

One difficulty in evaluating the FNN approach is that once trained, the models cannot be applied to another dataset (such
as that from an intercomparison project) in the likely case that it uses a different vertical grid, since the number of inputs and
outputs are fixed and correspond to collapsed and concatenated vertical profiles, and may be large depending on the vertical
resolution and the number of gases considered. The high dimensionality in turn implies a difficulty for the models to generalize
to e.g. warmer and moister climates given the "curse of dimensionality". Moreover, to obtain reasonable heating rates, the
typical approach has been to predict profiles of heating rates in addition to fluxes at the top-of-atmosphere and surface, as
opposed to predicting fluxes and computing heating rates physically. The downside of this approach is that it breaks energy
conservation as the fluxes and heating rates are likely to be inconsistent.



An alternative which offers better accuracy, reliability and interpretability at the cost of a smaller speed-up is to use ML only to predict optical properties and couple it to a radiative transfer solver. This may be justified by considering that radiation schemes solve the radiative transfer equation (using the two-stream approximation) to obtain accurate estimates of broadband
radiative fluxes, after having first computed spectral optical properties of gases, clouds and aerosols, ideally in separate modules (Fig. 1, Hogan and Bozzo, 2018). In contrast, parameterizations of other processes are often based more on empirical relationships and simplified theories (Wang et al., 2022). The computation of optical properties (which control the absorption, emission and scattering of radiation) within radiation schemes are likewise data-driven, as they rely on look-up-tables. This is arguably a much easier and more suitable problem for neural networks than computing radiative flows from one vertical level
to the next, especially for feed-forward networks which do not structurally incorporate the vertical dependencies of the latter.

FNNs were consequently developed to emulate the gas optics parameterization RRTM for General circulation model applications—Parallel (RRTMGP) (Pincus et al., 2019) in two different studies, which found speed-ups of 2–6x compared to the original code (Ukkonen et al., 2020; Veerman et al., 2021). In the former study the NN gas optics was combined with a refactored radiative transfer solver to speed up the entire radiation scheme (without cloud or aerosol optics) by a factor of
1.8–3.5. Recently, Ukkonen (2022) compared different emulation strategies for shortwave radiation, and found that using NNs for gas optics hardly affected accuracy at all, whereas replacing the entire scheme with FNNs was the fastest but also least accurate approach, with heating rates computed from fluxes having a RMSE of 1.35 K day$^{-1}$. An interesting middle-ground was found in using bidirectional recurrent neural networks, which structurally resemble physical computations, to emulate the full radiation scheme. This produced far more accurate fluxes (RMSE of 1–1.5 W$^{-2}$) and heating rates (RMSE 0.16 K day$^{-1}$)
than FNNs while offering a smaller, but still appreciable speedup.

While the recent studies show that gas optics emulation is at this point more suitable for operational implementation than replacing the full radiation scheme, not least due to inherently better generalization (e.g. to arbitrary vertical grids), the evaluations were based on offline radiation computations (Ukkonen et al., 2020; Veerman et al., 2021; Ukkonen, 2022). As the offline accuracy does not necessarily reflect its performance when coupled with other components (Liu et al., 2020), the NN version of
RRTMGP (Ukkonen et al., 2020) is in this work integrated into the ecRad radiation scheme used in the Integrated Forecasting System (IFS), which is a global numerical weather prediction model developed at the European Centre for Medium-Range Weather Forecasts (ECMWF). New NN models are trained on reduced-resolution RRTMGP $k$-distributions that recently became available, which have around the same number of $k$-terms as the older RRTMG scheme that is used operationally in the IFS. Free-running simulations are then performed in order to test the generalization and accuracy of the NNs in a prognostic
setting. In the context of wider literature on ML-based parameterizations, our focus on gas optics represents a more typical model development approach focusing on modules, and aims for immediate application in numerical weather prediction (NWP) and climate models. On the other hand, the gas optics scheme plays a very central role in climate models as pointed out by Hogan and Matricardi (2022) and is often computationally a significant part of the full radiation scheme (Hogan and Bozzo, 2018; Ukkonen et al., 2020).
The structure of the paper is as follows: Section 2 briefly describes the ecRad and RRTMGP-NN codes, and the implementation of RRTMGP-NN in ecRad. Section 3 provides an overview of the machine learning methodology, which has been refined



to reproduce radiative forcings with respect to individual gases more accurately. The results are then presented in Section 4, consisting of an offline evaluation, and a prognostic evaluation where the impact of the new gas optics schemes (RRTMGP and RRTMGP-NN) on model climate is determined using a small ensemble of year-long IFS simulations.

## 2 Codes

### 2.1 ecRad

ecRad is a radiation scheme developed at ECMWF and used operationally in the IFS since 2017 (Hogan and Bozzo, 2018). It is highly configurable with multiple options for gas optics, cloud optics, aerosol optics, and radiative transfer solvers which represent cloud heterogeneity in different ways and support various cloud overlap assumptions.

### 2.2 RRTMGP

RRTMGP is a recent gas optics scheme with a correlated $k$-distribution that is based on state-of-the-art spectroscopy. It it part of a freely available toolbox, RTE+RRTMGP, that couples it to a radiative transfer solver (RTE). The radiation scheme seeks to balance accuracy, efficiency and flexibility and both the code and data continue to evolve (Pincus et al., 2019). The original RRTMGP $k$-distributions have a relatively large number of $k$-terms, also known as $g$-points: 224 in the shortwave (SW) and 256 in the longwave (LW), corresponding to 16 in each SW and LW band, of which there are 14 and 16 respectively. Recently, reduced $k$-distributions with half the number of $g$-points (112/128) have been generated from the full distributions. This was done using the same approach as in the evolution from the RRTM scheme (Mlawer et al., 1997) to its reduced-resolution version designed for GCMs, RRTMG (Iacono et al., 2000), namely by iteratively combining neighboring $g$-points while attempting to minimize a cost function that includes fluxes, forcing, and heating rates (R. Pincus and E. Mlawer, personal communication, May 24, 2022). The reduction in $g$-points was similar in both cases; RRTMG, which is used operationally in the IFS, has 112/140 $g$-points.

As the number of $g$-points gives the number of pseudo-monochromatic radiative transfer calculations, it largely determines the cost of the whole radiation scheme. The NNs developed in this paper are therefore based on the new reduced $k$-distributions ("Reduced-RRTMGP"), and not the $k$-distributions with higher spectral resolution ("Full-RRTMGP").

### 2.3 RRTMGP-NN and implementation in ecRad

RRTMGP-NN is a neural network version of RRTMGP described in Ukkonen et al. (2020), available with a refactored version of the RTE solver which has columns as the outermost dimension (in terms of memory location) instead of $g$-points. The modified radiation scheme is referred to as RTE+RRTMGP-NN and like the original code, it is written in modern Fortran. The simple yet efficient NN kernel is based on BLAS routines for batched inference, exploiting the lack of vertical and horizontal dependencies in gas optics computations. (The code can also be run on GPUs, in this case NVIDIA cuBLAS is used for the matrix multiplications and OpenACC directives are wrapped around the remaining computations.) The underlying structure of



RRTMGP and RRTMGP-NN is loosely sketched out in Fig. 1 to demonstrate why the latter is more efficient. For the purpose of clarity, the computation of optical properties in RRTMGP has been simplified in the diagram: for each band, the code actually computes the absorption by 1-2 "major" gas species by 3D interpolation in temperature, pressure and $\eta$, relative abundance of the two major species (Pincus et al., 2019). The contribution from less important "minor" gases are computed separately using 2D interpolation for each minor gas in the band; therefore, a given band is written to several times. The NN gas optics instead predicts all spectral points simultaneously given an input matrix containing all gases.

RRTMGP-NN previously loaded models from ASCII files like the Neural-Fortran code (Curcic, 2019) that it is built upon. We have refined the code so that models are loaded from netCDF files, which contain not only the weights and activation functions, but also coefficients used for scaling inputs and outputs, as well as metadata about the training data. These files could in the future be expanded to replace the $k$-distribution files in their entirety, keeping relevant metadata and the look-up-table coefficients used to compute Planck sources from Planck fraction and temperature.

We now briefly describe the integration of RRTMGP into ecRad. The goal was to avoid larger changes in ecRad. However, since RTE+RRTMGP makes heavy use of Fortran derived types to specify e.g. gas concentrations and optical properties, use of existing RRTMGP interfaces would imply a significant amount of array copying to communicate between ecRad and RRTMGP derived types. Large changes in RRTMGP are not desirable either, because they reduce maintainability of the gas optics code.

With these conflicting goals in mind, a balance was sought with non-intrusive changes in both codes, but prioritizing minimal changes in ecRad.Firstly, the refactored radiation scheme with neural networks, RTE+RRTMGP-NN, was implemented instead of the reference gas optics code in order to make direct use of existing NN code. This has the advantage that RTE+RRTMGP-NN uses the same dimension order as ecRad with optical properties having $g$-points innermost and columns as the outermost dimension, removing the need for array transposes which can be a computational bottleneck (Ukkonen et al., 2020). While the NN fork of RTE+RRTMGP is currently only maintained by one person, the code is very similar to RTE+RRTMGP. The $k$-distributions are loaded from netCDF files which can be copied over as new ones are made available in the main repository.

The RRTMGP(-NN) package was then added as an ecRad subdirectory, with some small changes to remove the dependence on RTE. The source code of RRTMGP-NN is kept separate: it does not use any of the ecRad modules. Instead, new interfaces were written for RRTMGP-NN for easy interoperability with ecRad and avoiding array copying. For instance, the new interface for the longwave (*gas_optics_int_ecRad*) replaces the derived type arguments of the original RRTMGP interface, containing optical properties and Planck sources, with explicit shape arrays used in ecRad. The computational kernels remain the same, as in original RRTMGP they do not use derived types. In ecRad, another interface then prepares the RRTMGP-NN gas concentrations (columns outermost) by transposing the ecRad gases (columns innermost) and calls *gas_optics_int_ecRad* (longwave) and *gas_optics_ext_ecRad* (shortwave). ecRad has similar interfaces to the RRTMG and ecCKD (Hogan and Matricardi, 2022) gas optics schemes. The overhead from transposing the gases and thermodynamic arrays is very small. The $k$-distributions and NN models are stored in the ecRad "config" derived type in order to avoid introducing new arguments to the main interface.



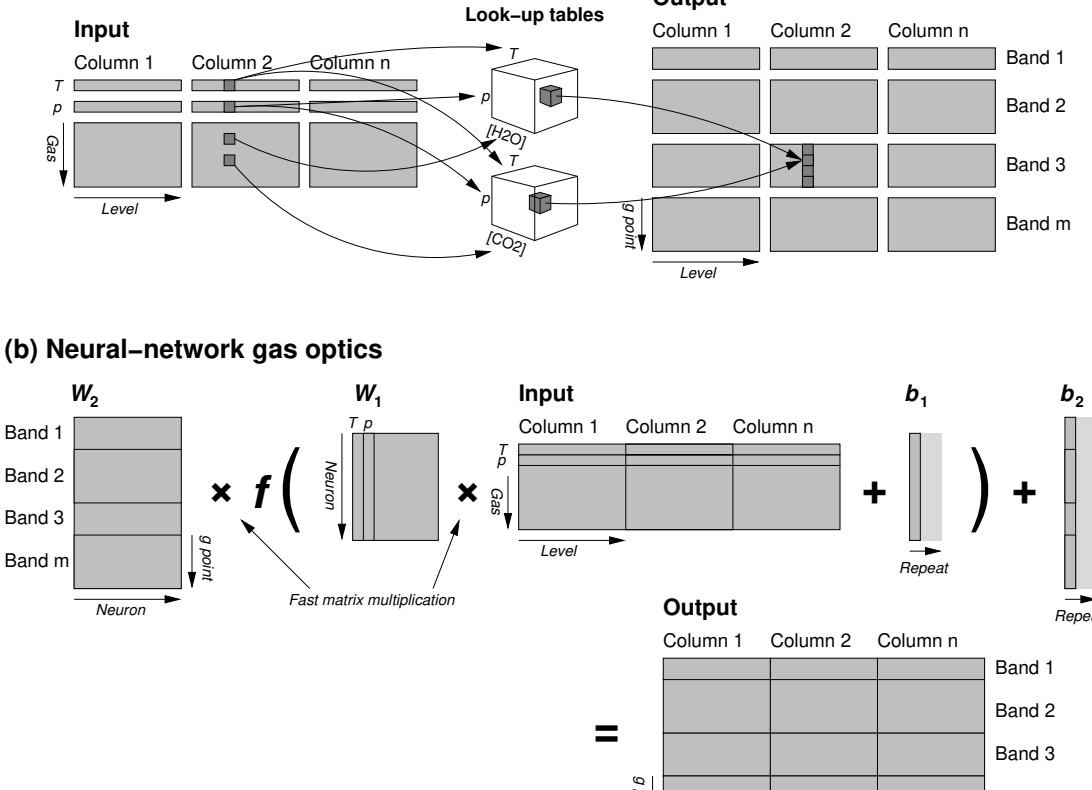

**Figure 1.** Schematic illustrating the data flow in a structure of (a) a conventional gas-optics scheme such as RRTMGP, and (b) RRTMGP-NN. For a given band, RRTMGP makes several calls to interpolation kernels to compute gas-specific contributions to optical properties, accounting for temperature ($T$) and pressure ($p$) dependencies, as well overlap of major gases in the band. These kernels loop over the $g$-points in a single band, of which there are only 1-12 in the reduced $k$-distributions, leading to poor vectorization. The steps are repeated for each vertical level and column. The NN (b) achieves better performance i) by predicting a vector containing all $g$-points from a vector containing $T$, $p$, and the mixing ratios of all gases (treating gas interactions implicitly by the NN), and ii) batching the computations for multiple atmospheric levels and columns by expressing the core NN computations as matrix-matrix multiplications (between weights $\mathbf{W}$ and input matrices) that are delegated to a BLAS library. For each hidden NN layer, this is followed by addition of biases $\mathbf{b}$ and transformation by a non-linear activation function $f(x)$. The schematic shows a network with only one hidden layer; in this study two hidden layers are used.

## 3 Machine learning

155 In training NNs to emulate RRTMGP, we use a similar methodology as in Ukkonen et al. (2020), where detailed offline evaluation against line-by-line computations suggested a similar level of accuracy in overall fluxes and heating rates as the





original scheme, despite using fairly simple NN models with two hidden layers and 16-48 neurons in each hidden layer. Various aspects of the methodology are refined from the previous paper as described below.

## 3.1 Data

We use similar training data as in Ukkonen et al. (2020), in which a diverse and extensive data set was prepared from several sources, including atmospheric profiles used in previous radiation studies, as well as data from future climate experiments and a reanalysis. These initial data sets were synthetically supplemented, or extended, by varying greenhouse gas concentrations both manually and by using Hypercube sampling. The data in this study differs from Ukkonen et al. (2020) principally in two ways. Firstly, data provided by the Radiative Forcing Model Intercomparison Project (RFMIP, Pincus et al., 2016), comprising of 100

profiles and 18 perturbation experiments, now serves as an independent validation dataset used for early-stopping (section 3.3) instead of training. These profiles were designed to assess global mean clear-sky errors in instantaneous radiative forcing and should be well-suited as an out-of-sample test for our purposes. Secondly, a different dataset based on the CAMS reanalysis (Inness et al., 2019) is used. The new CAMS data uses the same approach as the IFS and the Correlated $k$-distribution Model Intercomparison Project (CKDMIP, Hogan and Matricardi, 2020), where only nine gases are considered, but the radiative

forcing of many minor greenhouse gases is represented by artificially increasing the concentration of CFC-11 (Meinshausen et al., 2017). The height dependence of these gases is represented, and other RRTMGP gases are set to zero. (Neither of these generally applies to the remaining training data, where all minor RRTMGP gases are included, but as scalar concentrations).

The reanalysis profiles are designed to encompass the variability in present-day atmospheric conditions, with the following steps taken to increase variance and capture extremes. Starting from an initial pool of roughly 164 000 profiles spanning global

reanalysis data from 2008 and 2017 and interpolated to a 320 km resolution equal-area grid (Ukkonen, 2022), 1000 profiles were drawn. Of these, 17 were selected to contain the minimum and maximum of temperature, humidity and ozone at different pressure levels (a total of 9 variables) in the whole dataset, similarly to Hogan and Matricardi (2020). Another 486 profiles were selected by constructing $k = 81$ k-means clusters which are clustered in the 9 dimensions represented by the variables in the previous step. From each cluster, which the $k$-means algorithm ensures are as different to other clusters as possible, 6

random profiles were selected. The remaining roughly 500 profiles were randomly drawn from the entire dataset minus ones already chosen. Vertical profiles selected by the minimum-maximum, semi-random and random method are depicted in Fig. 2.

The 1000 CAMS profiles were then expanded into 42 experiments or scenarios where $CH_4$, $N_2O$, CFC11-eq and CFC12 are varied similarly to Hogan and Matricardi (2020). The $1000 \times 42 \times 60$ (layers) $\approx 2.5$ million samples make up roughly 47% of the 5.42 million total training samples. The remaining parts comprise i) end-of-century CMIP6 data corresponding

to a high-emissions scenario, ii) profiles from the "mean-maximum-minimum" CKDMIP dataset, and iii) 42 profiles used for tuning RRTMGP; all of which were expanded into up to hundreds of experiments as described in Ukkonen et al. (2020).

## 3.2 Choice of inputs and outputs

Our RRTMGP emulator predicts layer-wise optical properties from an input vector which contains gas mixing ratios, temperature, and log-pressure. The NNs take as input all the RRTMGP gases and output all $g$-points, which results in better





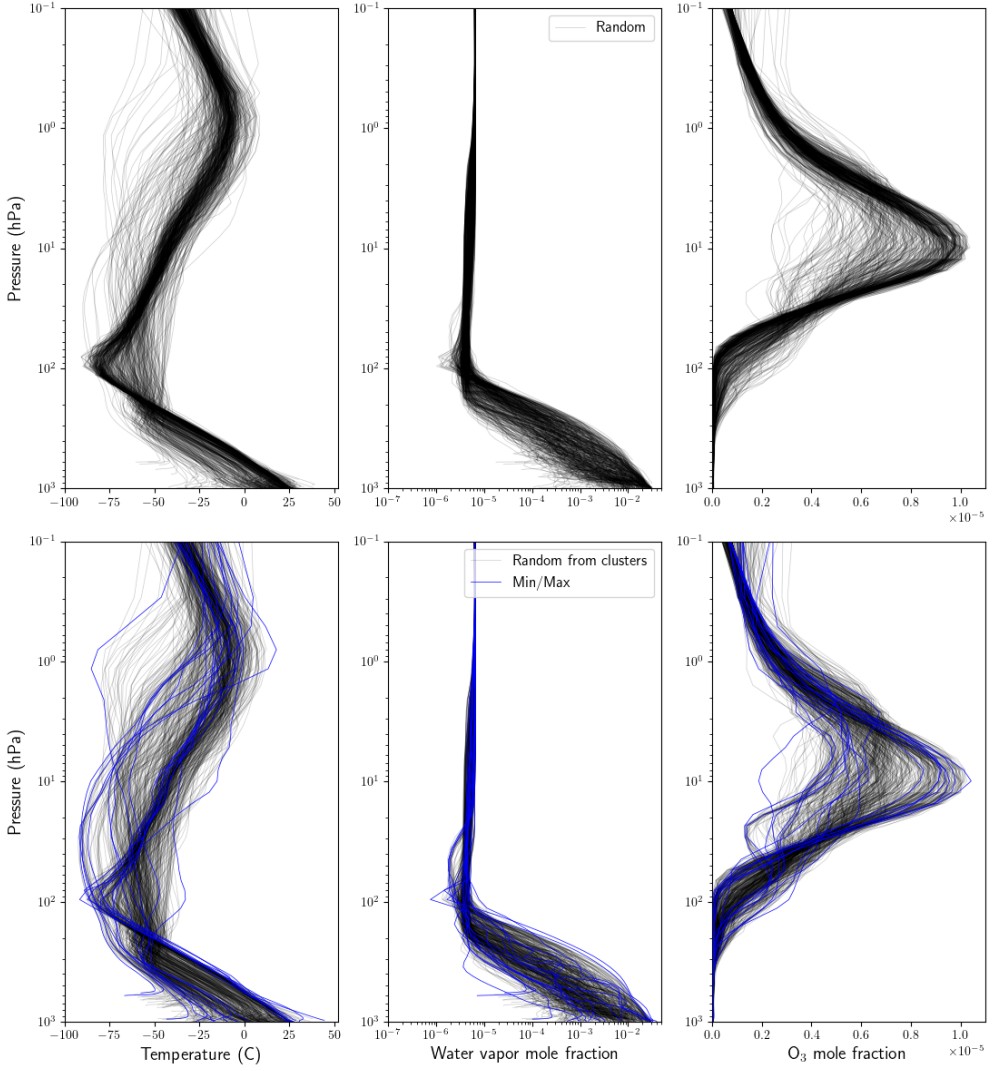

**Figure 2.** Vertical profiles of temperature, water vapor and ozone selected from the CAMS data as described in Sect. 3.1. The top panel shows 486 random profiles (black), and the bottom panel shows 486 profiles drawn from k-means clusters (black) and 17 that were selected to sample minimum and maximum values (blue).

computational intensity and efficiency than computing one band at a time, and the contributions from minor gases one gas at a time, as is done in the look-up-table (LUT) kernels in RRTMGP (Ukkonen et al., 2020). In the shortwave (SW), the NN outputs are absorption and Rayleigh cross-sections, while the longwave (LW) predictands are absorption cross-section and Planck fraction. Here, cross-sections refers to optical depth divided by the number of dry air molecules in a layer $N$. This allows generalization to arbitrary vertical grids, since optical depths are obtained in a separate step by multiplying the cross-sections with $N$. Meanwhile, *Planck fraction* is the fraction of a band's total Planck function that is associated with each LW $g$-point,



obtained by 3D interpolation in the original code. Like in RRTMGP, this is multiplied with the band-integrated Planck function at a level or layer (interpolated from a LUT using the temperature of that level/layer) to get the Planck function for each LW $g$-point. This retains a small LUT interpolation, but simplifies the NN model by requiring only $ng$ outputs, instead of $3 \times ng$ to directly predict the Planck functions used in reference RRTMGP, or $2 \times ng$ to get the Planck functions in RRTMGP-NN.

(The original code has one Planck variable for each layer and two for each layer interface, the upward and downward emission, whereas RTE+RRTMGP-NN has one for each layer and layer interface. ecRad only uses one Planck function, defined at layer interfaces). Reducing the number of NN outputs can decrease model complexity and runtimes, since most of the floating point operations occur in the final NN layer given $N_{gpt} > N_{neurons} > N_{inputs}$. However, in this work a single LW model is used which predicts both absorption cross-sections and Planck fractions as one big vector. This may not be the fastest approach but

has the benefit of easing the optimization procedure described in the next section.

In addition to predicting cross-sections instead of optical depths, to obtain good results with less complex models it is useful to pre-process both inputs and outputs. Specifically, square root transformations are used for all outputs and some inputs to make their distributions more uniform, and afterwards the inputs are scaled to the 0-1 range and outputs are scaled to have roughly zero mean and unit variance by using a variant of standardization that preserves correlations between different outputs

(Ukkonen et al., 2020).

### 3.3 Optimizing for fluxes when predicting optical properties

Training gas optics ML models presents a tuning challenge, as the variables we ultimately care about are not optical properties but radiative fluxes and heating rates - outputs from the solver. We previously found it relatively easy to develop gas optics NNs which upon implementation in the radiation code result in low mean errors in fluxes and heating rates, but difficult to obtain

accurate radiative forcings at the top-of-atmosphere or surface with respect to changes in the concentration of individual gases, especially minor ones (Ukkonen et al., 2020). The problem is likely to stem from predicting aggregated optical properties, instead of computing the contribution from minor gases separately (as is usually done in $k$-distributions), which is more efficient but leads to major gases dominating the loss function. Mostly accurate radiative forcings for CKDMIP gases were ultimately obtained via a time-consuming, iterative process where new models were continuously trained, evaluated, and the training data

expanded. In this work we have attempted to automate the optimization with regards to fluxes, heating rates and forcings to at least some extent by adding two new techniques to the training methodology.

Firstly, errors in fluxes and heating rates were monitored during training. While these metrics cannot easily be used for optimizing the NN weights, they can be used as a criteria to know when to stop training (*early-stopping*), or to optimize NN hyperparameters. Therefore, a Python training program was written where at the end of every epoch, the NN models are saved

to a file, and the Fortran radiation program is called with the new model, passing the location as a command-line-argument. The Fortran program runs RTE+RRTMGP-NN on a validation dataset, and writes some error metrics to standard output, which are finally read by the training program. For validation we used the RFMIP dataset consisting of 100 profiles and 18 different perturbation experiments, which allowed computing radiative forcing errors with respect to $CH_4$, $N_2O$, and total forcing errors with respect to all RRTMGP gases. In addition, a benchmark line-by-line solution was available for this data, meaning that the





total (parameterization) error can be determined instead of only emulation (NN) error. Our goal was to develop NNs that have a similar level of accuracy as RRTMGP; that is, emulation errors should be smaller than parameterization errors. The error metrics were normalized by the RRTMGP values, so that a value of one indicates the same level of performance as RRTMGP, and larger values indicate worse performance. An overall "radiation error" was computed by taking the RMS value of a total of 8 metrics which differ slightly for the longwave and shortwave (Table 1). This overall metric was used in the early stopping

criteria and the model weights from the best epoch (a minimum in the metric) were saved.

**Table 1.** Metrics that comprise the overall "radiation error". TOA = top of atmosphere, IRF = instantaneous radiative forcing, "future-all" = a radiative forcing experiment with perturbed atmospheric conditions in addition to greenhouse gas concentrations.

| Metric | Longwave | Shortwave |
|---|---|---|
| MAE Heating rate | X | X |
| MAE Heating rate (present-day) | X | X |
| MAE Heating rate (preindustrial) | | X |
| MAE Heating rate ("future-all") | | X |
| Bias surface downwelling flux | | X |
| Bias TOA upwelling flux | X | |
| Bias TOA IRF (present-day - preindustrial) | X | |
| Bias TOA IRF (future - present-day) | X | |
| Bias TOA IRF (future - preindustrial) | | X |
| Bias surface IRF (future - preindustrial) | X | X |
| Bias surface IRF CH4 (present-day - preindustrial) | X | X |
| Bias surface IRF N2O (present-day - preindustrial) | X | |

Secondly, a custom loss function was devised to minimize the error in the difference in $y$ associated with different perturbation experiments, in addition to mean-squared-error of $y$, where $y$ are the scaled NN outputs. The new loss function indirectly measures radiative forcing errors (albeit weakly due to a non-linear dependence between spectral optical properties and broadband fluxes) and has the form:

$$loss = \alpha \sum_{i=1}^{N} (y_i - \hat{y}_i)^2 + (1-\alpha) \sum_{\substack{i=1 \\ i\,\text{odd}}}^{N} \Big( (y_{i+1} - y_i) - (\hat{y}_{i+1} - \hat{y}_i) \Big)^2,$$

where $y$ and $\hat{y}$ are the target and NN output vectors, respectively, and $\alpha$ is a coefficient that was set to 0.6 for LW and 0.2 for SW after some testing. The second term measures the error in the difference in $y$ between different perturbation experiments if the data is organized so that adjacent samples (of a total $N$ training samples) correspond to different experiments but the same columns and vertical layers, which was achieved by transposing the data so that the experiment dimension is innermost.

In addition, the experiments should be designed so that every odd element and its neighbour relate to the goal, which was minimizing the TOA and surface forcing errors of individual gases. Therefore, RFMIP-style experiments such as present-day




versus future concentrations of all greenhouse gases, or 8X $CO_2$ versus preindustrial $CO_2$, should be avoided, as they can easily dominate the error compared to varying the concentration of minor greenhouse gases (which was the challenge to begin with). This requirement was only partially fulfilled since we wanted to make use of existing data. Though rather convoluted, and requiring bespoke data, the approach does reduce forcing errors (Fig. 3).


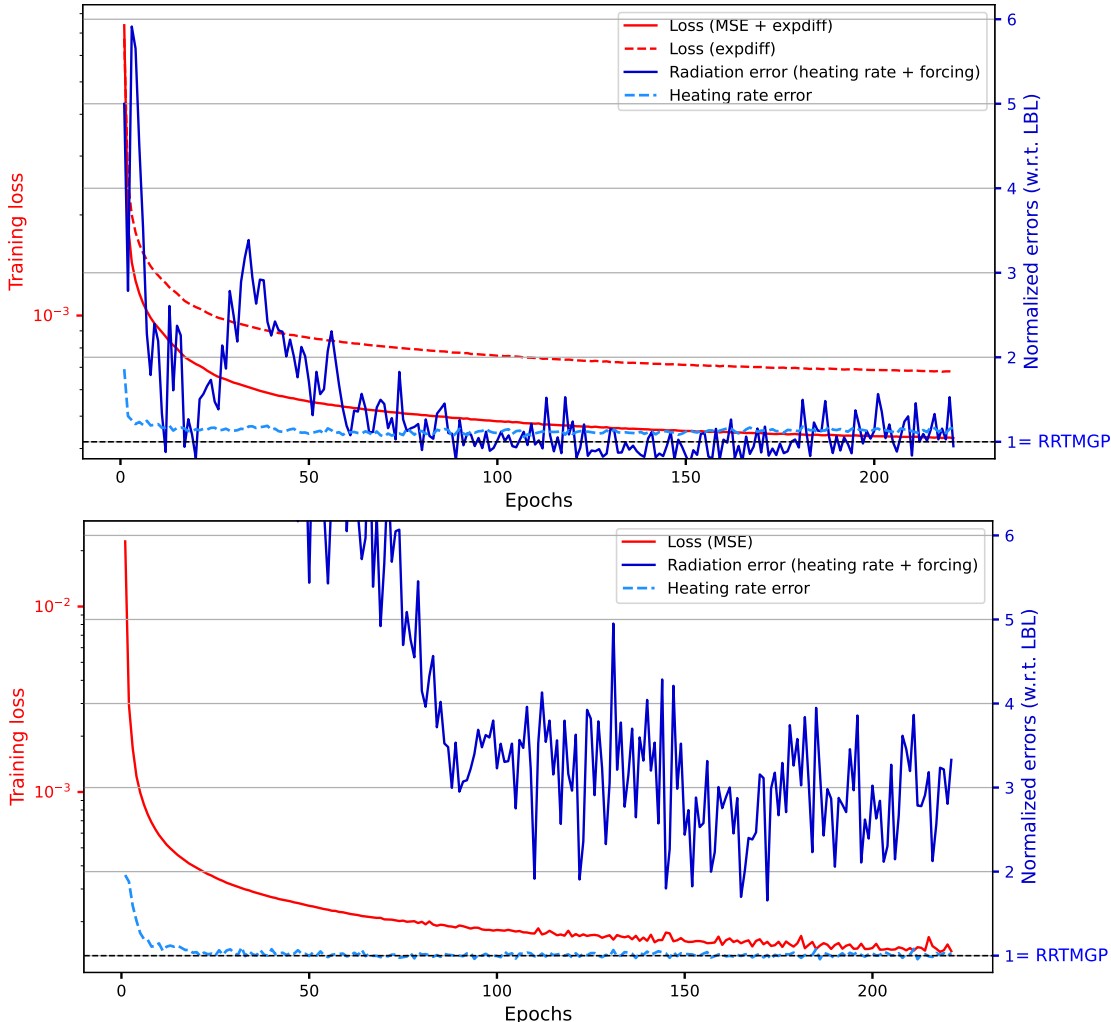

**Figure 3.** Monitoring of heating rate error (dashed cyan line, given by the mean of the heating rate metrics in Table 1), and the total radiation error (solid blue line, given by the RMS of the metrics listed in Table 1) when training the final LW gas optics model using a hybrid loss function and early stopping (top), and training for the same number of epochs with a regular loss function (bottom). The monitored errors (right axis) are computed using the RFMIP data with respect to line-by-line results and normalized by the RRTMGP value. Also shown are the training losses (solid and dashed red lines). The larger radiation error when not using the hybrid loss function (b) was largely due to a single metric, the surface radiative forcing of N2O (not shown).



In the end, there was still a substantial random element in obtaining good results, and several models were trained before settling on the final models (based on errors with respect to training data, and not the independent offline evaluation, which was only performed once). To obtain a satisfactory LW model the early-stopping criteria was loosened to 70 consecutive epochs of no improvement. In addition, increasing the number of hidden neurons from Ukkonen et al. (2020) seemed to improve results.

The final LW model has 64 neurons in two hidden layers, and the SW models have 32 neurons in two hidden layers. All models use the "softsign" activation function and were trained using the Adam optimizer, a batch size of 2048 and learning rate of 0.01.

Future studies could explore directly minimizing flux and forcing errors when training NN-based gas optics models. Doing this via gradient descent optimization would require differentiating the radiative transfer solver to obtain the derivative of fluxes

with respect to changes in optical properties (and NN weights), which should be possible using automatic differentiation tools like the Python library JAX (Bradbury et al., 2018) if the radiative transfer code was re-written in the supported language or API (JAX, for instance, has an API based on NumPy).

## 4 Results

In this section we evaluate the accuracy and speed of ecRad with different gas optics schemes (RRTMGP, RRTMGP-NN, and

the older RRTMG scheme) in both an offline and online setting. The results were obtained using an optimized development version of ecRad which refactors the Tripleclouds (Shonk and Hogan, 2008) and SPARTACUS (Hogan et al., 2016) solvers for better efficiency and includes the new RRTMGP(-NN) gas optics. One change is that reflectances and transmittances are computed in the same numerical precision as the rest of the model (in the operational version of ecRad, these two-stream computations are always performed in double precision), which improves the single-precision performance of all solvers in

ecRad. The updated solvers also improve efficiency by batching the reflectance-transmittance computations for multiple vertical levels. The optimizations, to be described in a forthcoming paper, have a negligible impact on fluxes and heating rates while making Tripleclouds significantly cheaper, and thus increase the share of the gas optics in ecRad's total runtime.

The prognostic evaluation and offline timings were obtained with an ecRad configuration close to operational IFS Cycle 47r3, which is similar to the 46r1 settings given in Table 2 of Hogan and Bozzo (2018), except for replacement of the pure

exponential cloud overlap assumption with "exponential-random" whereby an vertically contiguous cloud layers are partially correlated but cloud layers separated by clear sky are randomly overlapped. Furthermore, instead of the McICA solver we use Tripleclouds due to the latter being noise-free and a likely candidate for operational use in a near-future cycle.

It should be noted that the RRTMGP results were not produced using the original RTE+RRTMGP package, which uses two Planck source functions for half-levels which are then combined into one, and the LW results (Figures A1-A4 in Appendix

A), and could be very slightly impacted by the simpler computation of Planck source in RRTMGP-NN. For simplicity the RRTMGP-NN code configured with look-up-tables and not NNs is hereafter referred to as RRTMGP.



## 4.1 Speed-up

The runtime of ecRad with different gas optics schemes was evaluated offline using 10,000 input profiles that were randomly sampled from a global snapshot saved from a high-resolution IFS run. Figure 4 shows timing results obtained on a single node of the new ECMWF AMD-based supercomputer in Bologna, to which the migration of ECMWF's operational forecast is expected later in 2022.

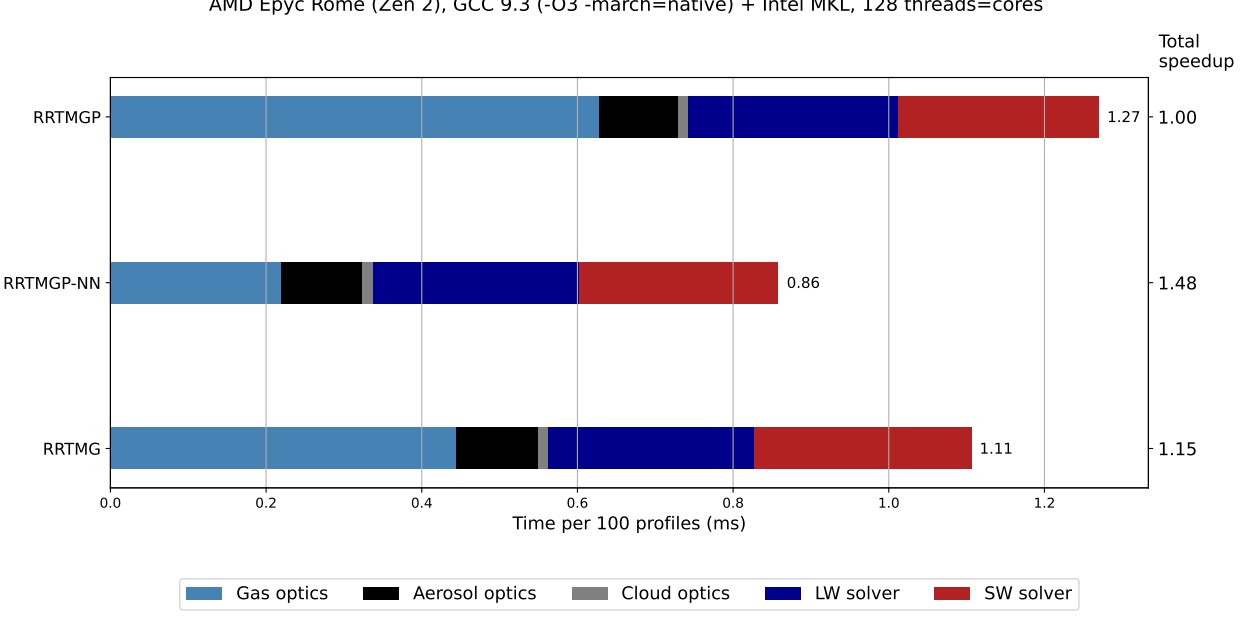

**Figure 4.** Runtime of ecRad in single precision per 100 atmospheric profiles with 128 levels (x-axis) when combining different gas optics (y-axis) with the TripleClouds solver, broken down into components. Three gas optics schemes are compared: RRTMGP with reduced spectral resolution (112 SW and 128 LW $g$-points), its neural network version (RRTMGP-NN), and the older RRTMG scheme with 112 (SW) and 140 (LW) $g$-points. The 10,000 original columns with 137 vertical levels were repeated 4 times into 40,000 columns that were blocked in an OpenMP loop in which ecRad is called (blocking the derived type arguments and using a block size of 8 columns to reflect IFS use). Computations were repeated 10 times in an outermost loop, and furthermore the program was run 3 times, with the fastest result saved. The component runtimes are means of per-thread values reported by GPTL, but normalized to add up to the total time spent in the OpenMP loop (annotated to right of the bars), which was incrementally higher than the sum of section means. The speed-up relative to RRTMGP is shown to the right of the bar plot. Platform: 2x 64-core AMD EPYC 7H12 CPUs, GNU Fortran compiler version 9.3 and Intel MKL library 19.0.5 (used for general matrix-matrix multiplication (GEMM) in RRTMGP-NN).

With Reduced-RRTMGP, the runtime of ecRad is slightly higher than with RRTMG due to the new gas optics scheme (shown in light blue) being more expensive. The older gas optics module is faster by a factor of 1.41 despite the similar spectral resolution. The poor efficiency of RRTMGP is explained by short inner loops in the LUT code that iterate over the number of $g$-points in a band (which is only 1-12 for the smaller $k$-distributions), leading to poor vectorization. However,





speed is improved drastically with the NN version of Reduced-RRTMGP, which is roughly 2.87 times faster than the LUT-based code, and 2 times faster than the operationally used RRMTG scheme. The resulting speedup in the full radiation scheme is 1.48x and 1.29x compared to RRTMGP and RRTMG respectively.

We note that the speed-up achieved with RRTMGP-NN is heavily influenced by many factors, such as the hardware and software platform (especially the performance of the BLAS library) but also the number of gases included. While our new RRTMGP-NN models support all of the 11 minor greenhouse gas species in RRTMGP, here only the reduced set of gases (with CFC-11-equivalent) is used as input in ecRad and evaluated (remaining gases were set to zero in the NN inputs). Including a large number of greenhouse gases would slow down reference RRTMGP where the kernel has a loop over minor gas species, whereas the runtime of RRTMGP-NN would hardly be affected, improving the speed-up further. Indeed, one benefit of our NN approach is that it can account for many minor gases essentially for free in a computational sense (although, not necessarily accurately, and in the next section we do not evaluate radiative forcings with respect to non-IFS gases).

### 4.2 Offline evaluation

Independent validation of the NN gas optics models was carried out by using data and tools from CKDMIP (Hogan and Matricardi, 2020). The data are from the 'Evaluation-1' dataset, which was not used for training. The accuracy of the new RRTMGP-NN SW model, relative to a line-by-line benchmark, is first shown in Fig. 5 for the present-day scenario. The NN has almost identical accuracy as the Reduced-RRTMGP scheme it was trained on (Fig. 6), particularly in terms of heating rates, which in both cases have an RMSE of only 0.056-0.057 K d$^{-1}$ below 4 hPa. Surprisingly, the bias and root-mean-square errors in upwelling fluxes are actually smaller when using NNs. Results using the other CKDMIP concentration scenarios (Glacial Maximum, Preindustrial, and Future) are not presented here but are very similar, with the NN gas optics resulting in better upwelling fluxes but similar heating rates.

In general, the close emulation of RRTMGP was already demonstrated in Ukkonen et al. (2020) and so the remaining results are not discussed in detail, but the LW results for present-day and future scenarios are provided in Appendix A (Figures A1-A4). The improvement in upwelling fluxes (but not heating rates) is even more pronounced in the longwave, with RRTMGP-NN resulting in 40-50% lower biases in TOA upwelling flux than the original scheme for the various concentration scenarios. The result is therefore probably not a fluke, but attributable to using early-stopping based on a line-by-line benchmark. The most notable difference to our earlier paper is that the top-of-atmosphere and surface forcings with respect to N$_2$O, CFC11 and CFC12 have been improved with the help of a hybrid loss function and are now excellent (Fig. 7). We note that the new Reduced-RRTMGP $k$-distributions with 112 (SW) and 128 (LW) $g$-points seem to trade only a little accuracy for nearly halving the cost of the radiation scheme, compared to the original $k$-distributions with almost double the spectral resolution (https://confluence.ecmwf.int/display/CKDMIP). The only major degradation is in longwave heating rates in the mesosphere, with reduced-RRTMGP introducing a considerable warming bias in the mid-mesosphere.







**Figure 5.** Evaluation of Reduced-RRTMGP-NN shortwave fluxes and heating rates using the 50 independent profiles of the CKDMIP Evaluation-1 dataset with present-day concentrations of greenhouse gases. The left column (a, d, g) shows the reference profiles of upwelling flux, downwelling flux and heating rate from LBL calculations with five different values of the cosine of the solar zenith angle, $\mu_0$ (0.1, 0.3, 0.5, 0.7 and 0.9). The middle column (b, e, h) shows the corresponding biases (solid lines) and 95th percentile of errors (shaded area) using all 250 data points. The right column (c, f) depicts instantaneous errors in upwelling TOA and downwelling surface fluxes with the clusters corresponding to the different solar zenith angles.







**Figure 6.** As in Fig. 5 but for Reduced-RRTMGP.

## 4.3 Prognostic evaluation

We now describe results from a prognostic evaluation of RRTMGP-NN and RRTMGP using 1-year free-running simulations
with the IFS model. The model simulations consisted of four atmosphere-ocean coupled simulations 13 months long initialized
on 1 August of the years 2000, 2001, 2002 and 2003. After a 1-month spin-up for each simulation, the remaining 12 months
were averaged over each simulation. This configuration is very similar to that used in section 5 of Hogan and Bozzo (2018)
to evaluate the impact of changes to the radiation scheme; the simulations are long enough to capture fast atmospheric and




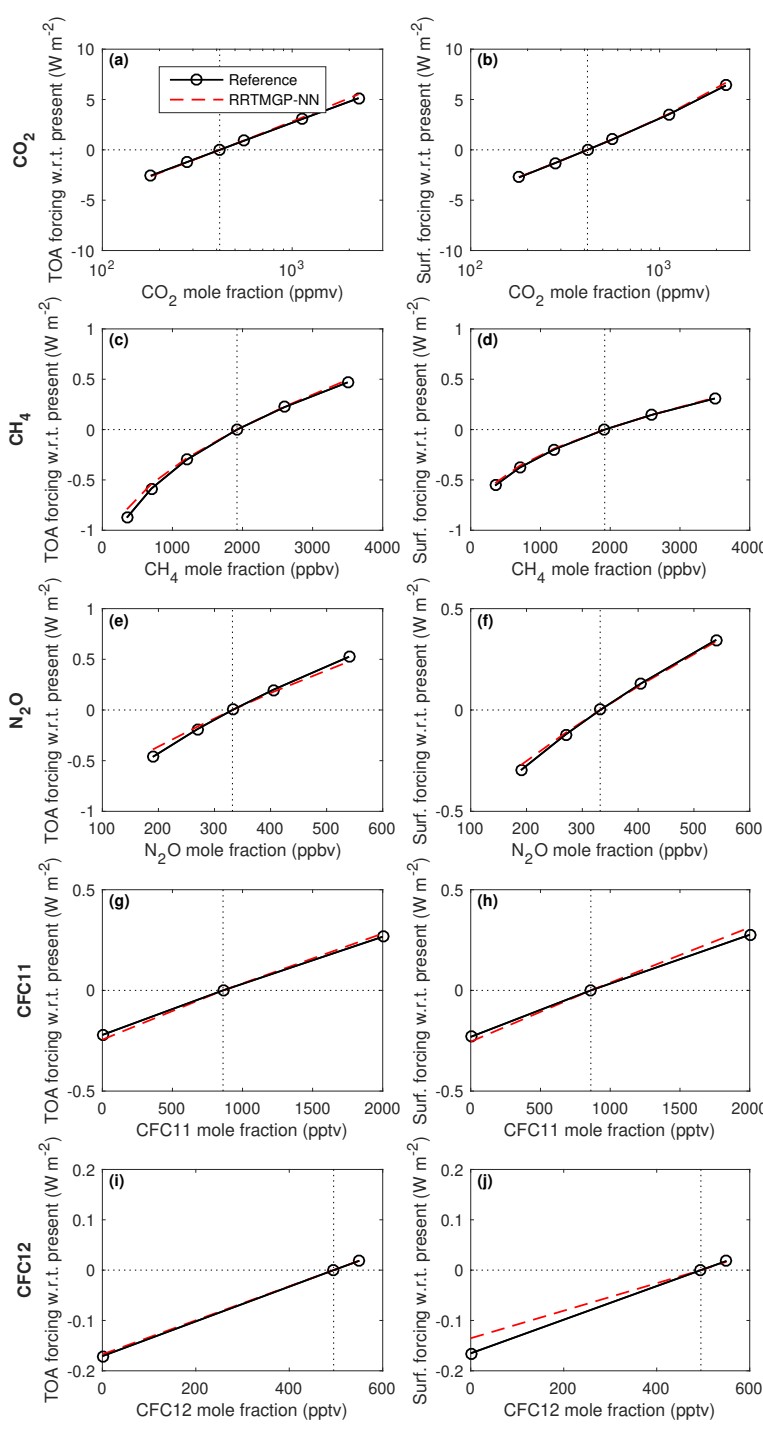

**Figure 7.** Comparison of RRTMGP-NN and LBL calculations of instantaneous longwave clear-sky radiative forcing at top of atmosphere (left column) and surface (right column) when perturbing different greenhouse gases (rows), averaged over the 50 Evaluation 1 profiles.




land-surface processes that respond to changes in the treatment of radiative transfer, but short enough that the response is not significantly affected by the longer-term changes to ocean circulation. The one-year forecast length also matches the longest

operational forecast length used in ECMWF's seasonal forecasts. The model configuration was as in operational IFS model cycle 47r3 except for the use of the Triplecluds rather than McICA solver. The horizontal resolution was $T_{Co}199$ (around 60 km) and 137 vertical levels were used. The radiation scheme was called every hour.

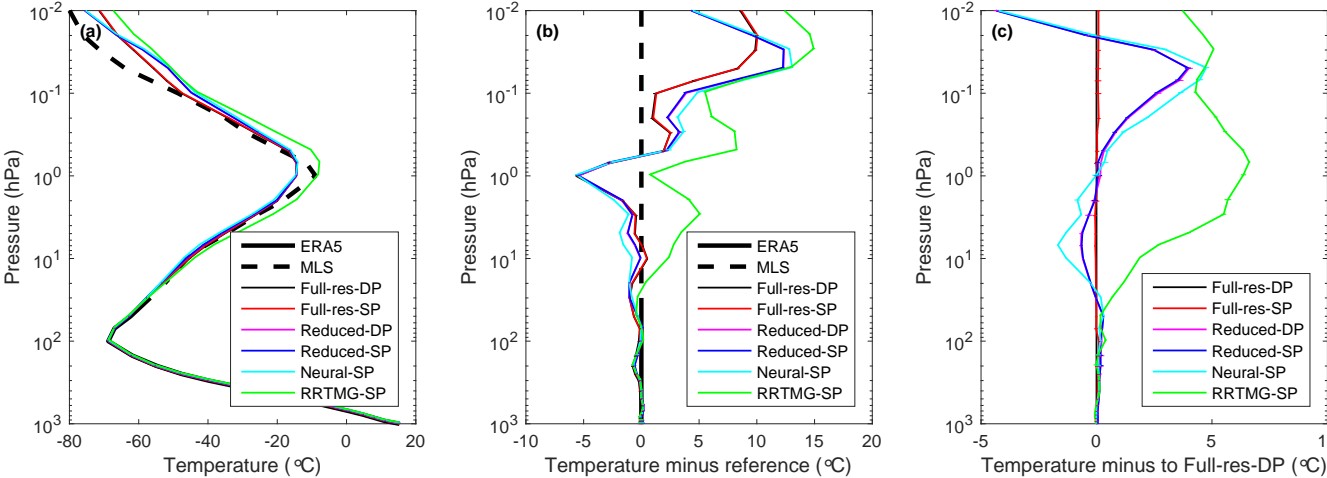

**Figure 8.** Evaluation of global/annual-mean temperature profiles from free-running simulations by the IFS. (a) Mean temperature, (b) difference against a reference dataset consisting of the MLS climatology above the 20-hPa height level and ERA5 below this level, and (c) difference against a simulation using full-resolution RRTMGP in double precision. The small horizontal bars give the 95% confidence interval as computed from differences between different years. As indicated in the legend, the simulations were performed in double or single precision (DP or SP) using the full- or reduced-resolution RRTMGP, the NN emulation of (reduced-)RRTMGP, or the older RRTMG gas optics scheme.

The impact of different gas optics schemes on annual-mean temperature from the surface to the lower mesosphere is shown in Fig. 8. Because RRTMGP is not, to our knowledge, routinely tested in single precision, both single and double precision

runs were performed with both Reduced-RRTMGP and Full-RRTMGP. The NN version of Reduced-RRTMGP was only tested in single precision (internally, RRTMGP-NN always uses SP, as higher numerical precision does not benefit NNs). In general, larger differences between the runs are only seen in the mesosphere and upper stratosphere, which are very sensitive to heating-rate differences. Comparison against a reference dataset based on the Microwave Limb Sounder (MLS) instrument above 20 hPa, and ERA5 reanalysis data below this level is depicted in Fig. 8 (b) and shows a 5-K warm bias in the upper stratosphere

for RRTMG, and larger in the mesosphere. The same bias was reported by Hogan et al. (2017), which they explained by the use of the older 'Kurucz' solar spectrum in the RRTMG version used in ecRad. RRTMGP uses a more recent solar spectrum with less ultraviolet radiation, resulting in closer agreement with MLS. In the mesosphere, RRTMGP-NN is much closer to the Reduced-RRTMGP scheme it is emulating than Reduced-RRTMGP is to Full-RRTMGP (Fig. 8 (c)). This strongly suggests that the emulation errors are small enough to be acceptable.





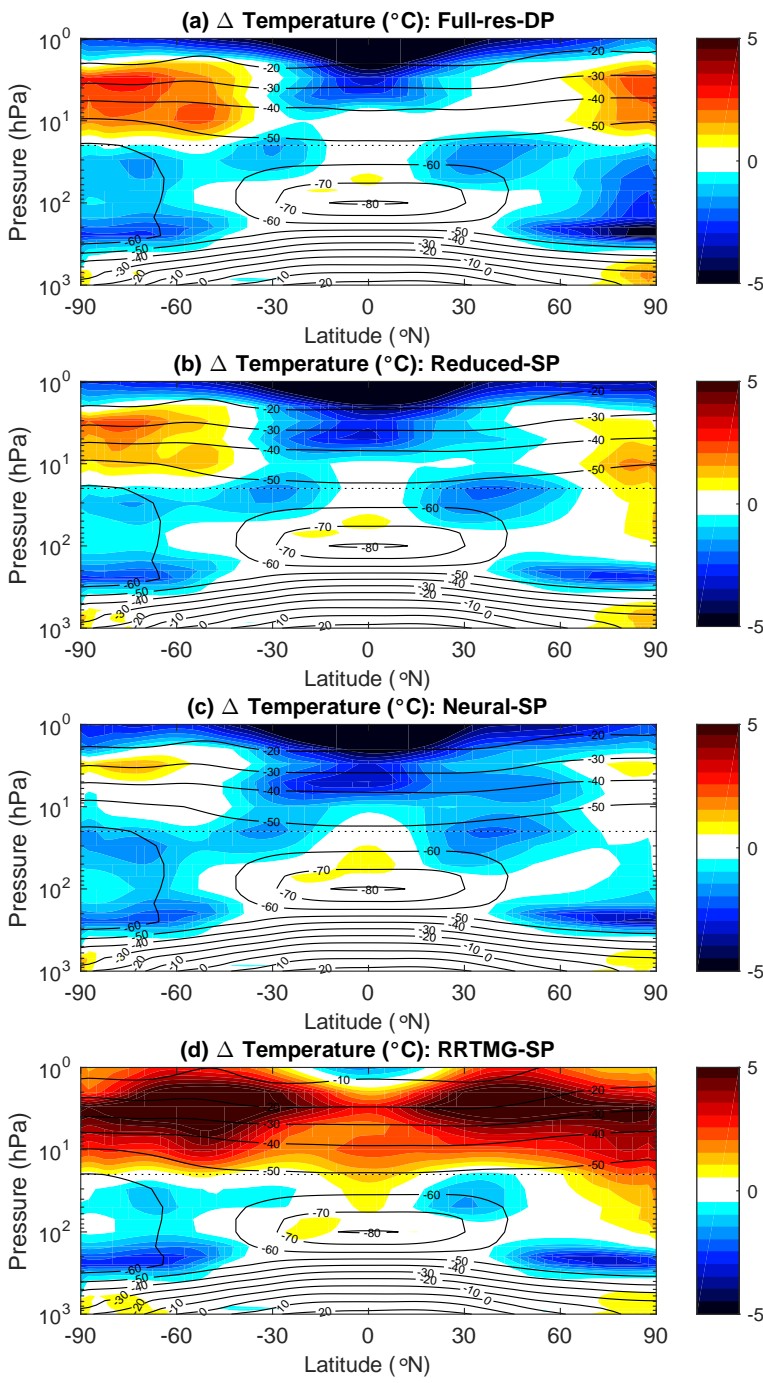

**Figure 9.** Similar to Fig. 8 but showing the the height-latitude cross section of mean temperature (black contours) and temperature difference (colors) against the reference datasets, and only until 1 hPa.



**Figure 10.** Zonal mean of different single-level quantities using the reference Full-RRTMGP run in double precision (a, c, e, g, i) and differences relative to this run (b, d, f, h, j) with the vertical lines indicating the 95% confidence interval: (a, b) 2-m temperature, (c, d) TOA net SW flux, (e, f) TOA net LW flux, (g, h) surface downwelling SW flux, and (i, j) surface downwelling LW flux.



A height-latitude cross section of temperature likewise shows larger differences between the old RRTMG scheme and RRT-
MGP than between different RRTMGP configurations and the NN version (Fig. 9). A strong warm bias in the stratosphere
is evident for RRTMG but less so for any version of RRTMGP, although the RRTMGP(-NN) runs do show a weaker upper-
stratospheric warm bias over high latitudes and a substantial cold bias in the tropical stratosphere.

Finally, Fig. 10 compares annual/zonal means of 2-m temperature, TOA net LW and SW fluxes, and downwelling SW
flux between simulations using different gas optics configurations. In general the differences relative to Full-RRTMGP are
statistically insignificant as the means fall within the error bars computed from the 4-year sample. The only clear exception
is RRTMG, which, for instance, at lower latitudes has significantly larger surface downwelling SW flux and smaller LW flux
than Full-RRTMGP. These findings are consistent in sign and approximate magnitude with the evaluation of RRTMG against
line-by-line calculations by Hogan and Matricardi (2020), although it should be stressed that these differences of 0–2 W m$^{-2}$
are still very modest.

## 5   Conclusions

In this paper we have evaluated RRTMGP-NN, a neural network version of the RRTMGP gas optics scheme, integrated into
ECMWF's radiation scheme ecRad, by performing both offline calculations and four 1-year simulations with the free-running
IFS model. Emulating only the gas optics component, instead of the full radiation scheme like in previous work, results in much
better accuracy. The NNs were trained on diverse datasets which cover a wide range of gas concentrations and atmospheric
conditions (including pre-industrial, present-day and future conditions). Although training on optical properties derived from
RRTMGP, we used early-stopping with respect to a validation set containing broadband fluxes computed by a line-by-line
model, based on metrics which included radiative forcings. Combining this with the use of a hybrid loss function, we were able
to reduce radiative forcing errors. Clear-sky shortwave and longwave fluxes, heating rates, and top-of-atmosphere and surface
forcings for 5 greenhouse gases were evaluated against line-by-line computations using an independent dataset spanning a
wide range of climate scenarios. In each case, any emulation errors introduced by RRTMGP-NN were virtually undetectable
compared to the parameterization error. Both schemes have heating rate errors within 0.056-0.057 K d$^{-1}$ (SW) and 0.102-0.116
K d$^{-1}$ (LW). Flux errors were generally below 1 W m$^{-2}$ and upwelling fluxes were actually improved by using RRTMGP-NN.
Fluxes for cloudy profiles would be even less affected as the treatment of clouds is identical in our approach.

The results from the online evaluation are also highly encouraging as they show very similar model climates for RRT-
MGP and RRTMGP-NN. Global/annual mean temperature profiles for the reduced-spectral-resolution RRTMGP and its NN
emulator are in closer agreement than different versions of RRTMGP are to each other, and all of these schemes, including
RRTMGP-NN, substantially reduce stratosphere and mesosphere temperature biases relative to the older RRTMG scheme. In
single-level fields, the differences between RRTMGP-NN and RRTMGP are obscured by natural variability. These results,
taken together with those from the offline evaluation, demonstrate that our RRTMGP-NN models are safe and suitable for
operational weather and climate models. To our knowledge, the same has not been demonstrated for full radiation scheme em-
ulators, whose substantial emulation errors particularly for cloudy profiles (Fig. 2, Song and Roh, 2021), and potential lack of





reliability and energy conservation, is especially worrisome for climate modeling applications. For instance, radiative forcings with respect to greenhouse gases, or cloud radiative effects, have not been evaluated in any emulator study we are aware of. We would also highlight the advantages of modular radiation schemes, which allows reducing model uncertainties in a continuous and interpretable manner by developing new and improved solvers, gas, aerosol or cloud optics schemes independently of one another.

Our evaluation of RRTMGP-NN was based on a setup with a reduced set of LW gases (using artificially increased CFC-11 concentrations to represent further gases); for applications where the individual radiative forcings of other RRTMGP minor greenhouse gases are of importance, such metrics can be evaluated offline and new NN models trained if needed. The data and tools are freely available and may be useful for future work. We encourage developers of new gas optics schemes to consider NNs as part of their toolbox, as they combine computational efficiency with algorithmic flexibility that allows avoiding structural assumptions, although training multi-gas models directly on line-by-line data could be challenging in terms of data generation and optimizing for minor gases.

In offline timings obtained on ECMWF's AMD-based supercomputer, using RRTMGP-NN instead of RRTMGP makes ecRad 1.5 times faster. This should be a significant reduction especially for climate models, which can spend a large share of the total model runtime on radiation. While studies emulating the entire radiation scheme have reported much larger speed-ups of 1–2 orders of magnitude, comparisons are often against old and slow radiation codes (e.g. Lagerquist et al., 2021), and gains may be much smaller against state-of-the-art schemes. For instance, by combining performance refactoring (that we plan to describe in a future paper) with algorithmic developments that allow reducing the spectral dimension (Hogan and Matricardi, 2022), the cost of ecRad with TripleClouds is reduced by an order of magnitude compared to the operational version using McICA (Hogan and Bozzo, 2018), which was already significantly faster than the previous ECMWF radiation scheme. Adopting common metrics for efficiency (e.g. time per column per CPU core) and accuracy, and standard datasets for independent verification, would no doubt be helpful for getting a clearer view of the usefulness of radiation emulators. Unfortunately, the latter may be difficult in practice given limitations such as inflexibility with regards to the vertical grid, which affect usual emulation approaches. Nevertheless, as machine learning does not offer a magic bullet to the accuracy/speed trade-off problem (Ukkonen, 2022), it is important that both are carefully evaluated in emulator studies.

*Code and data availability.* RTE+RRTMGP-NN is available on Github (https://github.com/peterukk/rte-rrtmgp-nn); the Fortran programs and Python scripts used for data generation and model training are found in the /examples/rrtmgp-nn-training subdirectory. The archived RTE-RRTMGP-NN 2.0 version and training data has also been uploaded to Zenodo (https://doi.org/10.5281/zenodo.6576680), as has the optimized non-official version of ecRad integrated with RRTMGP-NN 2.0 (https://doi.org/10.5281/zenodo.7148329).



## Appendix A: CKDMIP evaluation for the longwave

This section contains longwave results for the RRTMGP gas optics model with reduced spectral resolution (Reduced-RRTMGP), and its neural network (NN) version, using the experiment protocol, data and tools from the Correlated K-Distribution Model
Intercomparison Project (CKDMIP). Four figures are included, corresponding to different gas optics models and gas concentration scenarios: RRTMGP-NN, present-day (Fig. A1), RRTMGP, present-day (Fig. A2), RRTMGP-NN, future (Fig. A3), and RRTMGP, future (Fig. A4). The calculations were performed using a no-scattering solver with four discrete zenith angles in each hemisphere.

*Author contributions.* The conceptualization of the paper was mostly by PU. The machine learning work, including model design and data
generation, was carried out by PU. Implementation of RRTMGP-NN in ecRad was done by PU but with input from RJH. Both authors contributed to the offline evaluation (timings by PU). The IFS experiments were mostly designed by RJH and carried out by PU. PU prepared the manuscript with contributions from RJH.

*Competing interests.* No competing interests are present.

*Acknowledgements.* The authors thank Robert Pincus for reading the manuscript and providing suggestions to improve the clarity of the text.



**Figure A1.** Evaluation of Reduced-RRTMGP-NN longwave fluxes and heating rates using the 50 independent profiles of the CKDMIP Evaluation-1 dataset with present-day concentrations of greenhouse gases. Reference profiles of upwelling, downwelling flux and heating rate from LBL calculations, corresponding errors (b, e, h) with solid lines showing bias and the shaded area giving the 95th percentile of errors, and instantaneous errors in upwelling TOA and downwelling surface fluxes (c, f).







**Figure A2.** As in Fig. A1 but for the original Reduced-RRTMGP.







**Figure A3.** As in Fig. A1 but for the future scenario.





**Figure A4.** As in Fig. A1 but for the future scenario and Reduced-RRTMGP.





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
