# Peer review of "Implementation of a machine-learned gas optics parameterization in the ECMWF Integrated Forecasting System: RRTMGP-NN 2.0"

_EGUsphere, 2022_

## Author Response (AR2)

**RESPONSE TO REVIEWERS**

We thank both anonymous reviewers for providing helpful suggestions to improve the manuscript. These were mostly minor in nature and mainly concerned clarity of the text. We have tried to address the comments as best we could, improving clarity and adding a new table describing the training data (Table 1 in the revised paper). Specific responses to Referee 1 and 2 are found below on p1. and p6 respectively. Note that these were already posted on the discussion - the responses should be mostly identical here, except for the text alterations or additions for the revised manuscript, which we have bolded.

**REVIEWER 1**

*"..the manuscript feels a little incremental with respect to Ukkonen's previous RRTMGP NN paper. On itself, this is not a problem for GMD, but at the same time I think there is some opportunity to increase the novelty. I can assume that implementing all my suggestions would be too much to ask, but I do think that they might enhance the relevance of the paper."*

The novel aspects relative to the previous paper are mentioned in the abstract, they are:

- Prognostic evaluation of NN gas optics. While detailed offline evaluation could be considered by adequate by some, I believe many readers and potential users would want any NN-based parameterization to be evaluated using fully prognostic NWP or GCM simulations before seriously considering it in an operational context. Even when training on large and diverse data sets, given the high dimensionality of the gas optics problem, prognostic simulations are likely to include combinations of inputs that have not been seen during training.
- (ML method) Use of a hybrid loss function to (semi-) explicitly minimize radiative forcing errors with respect to changes in gas concentrations, which was a weakness of Ukkonen et al. 2020
- (ML method) Monitoring errors with respect to a line-by-line solution and early stopping with respect to these

In a revised version, we will more explicitly emphasize that the latter two are new additions: **"Using two new methods to improve accuracy - a hybrid loss function designed to reduce radiative forcing errors, and an early stopping method based on monitoring fluxes and heating rates with respect to a line-by-line benchmark - we train NN models on RRTMGP k-distributions with reduced spectral resolutions"**.

*"The paper feels a little outdated with the generation of the small g-point sets by Hogan and colleagues (as presented in ECMWF seminar). I could imagine that many people would prefer the small g-point set over the NN-optimized larger set. It would therefore be great if this study could also provide NNs for the small sets and show their performance with respect to the standard optical solver and the other g-point sets."*

The generation of small k-distributions by Hogan and colleagues does indeed represent a major development which arguably makes larger k-distributions outdated. However, the training and

evaluation of ecCKD in the Hogan and Matricardi (2022) was also based on a relatively small number of profiles, and it has yet to be shown that their gas optics schemes can match the accuracy of RRTMGP in all contexts. Indeed, IFS simulations have revealed a modest amount of random error in temperatures and velocities in the tropical troposphere that are believed to be due to the training being performed on too few atmospheric profiles. Work is ongoing to improve this, but RRTMGP is still a very relevant model in situations when accuracy is paramount. Regarding acceleration of ecCKD gas optics, the existing gas optics code based on look-up-tables is already fast, so NN emulation is unlikely to bring any significant reductions in total runtime.

*"Then, the reordering of the dimensions of the flux solver of RRTMGP is nice from a performance perspective, but at the same time it does not benefit the wider community if the baseline RTE+RRTMGP package comes without this flux solver. It would be a bit much to ask the authors to also generate NNs for this set up, but I think that the authors should have at least a discussion on this.*

*Lastly, the chosen dimension ordering is good for performance reasons, but it would not benefit any form of 3D, as such solvers (n-stream, ray tracer) most likely for memory reasons will have to loop over the gpt set as an outer dimension. To increase the durability of this work, it would therefore be great if the NN solver can deliver the optical properties in `(ncol, nrow, ngpt)` (Fortran ordering)."*

On the first point, "flux solver" is a bit confusing but the referee seems to refers to the fact that the version of RRTMGP implemented in ecRad uses g-points innermost, and so does its emulator, but official RRTMGP uses g-points outermost (internally, it used to have g-points innermost, that's now been changed). It would be possible to implement the NN inference code so that g-points (and gas concentrations) are outermost, without retraining any NNs, but it would likely be detrimental for performance, especially for ecRad. However, such a kernel is actually available in RTE+RRTMGP-NN and we mention this in the revised manuscript at the end of section 3.3.

Regarding the optimal dimension order, the section 2.3 in Hogan and Bozzo (2018), and appendix A of Ukkonen et al. (2020), give some arguments why g-points innermost should be better for performance: most notably, conditional operations depending on the presence of clouds and sun being above the horizon are a function of column but not g-point. We are also working on an ecRad optimization paper which shows that even with ecCKD, with only 32 g-points in the innermost loop, good vectorization can be achieved with some code restructuring which collapses the spectral and vertical dimensions whenever possible, leading to long inner loops (this would not be possible with columns innermost). While I understand the concerns about large memory footprints for 3D solvers using g-points innermost, I am confused how the 3D solver will achieve any SIMD vectorization if g-points are outermost? SIMD vectorization is a major source of performance on modern CPU's and if it's abandoned the performance will be poor (if this is indeed the case, I would be hesitant to declare that g-points innermost would not benefit 3D solvers).

**Small comments**

*- It would be good if the abstract specified explicitly what is new with respect to previous paper*

Now specifying this more explicitly in the abstract (see major reply).

*- The online validation method is not clear from the abstract*

I'm afraid this is not specific enough to address – what exactly is unclear? The last paragraph in the abstract describes what kind of simulations were performed and the most important conclusions from them.

*- Line 84: here the even smaller g-point sets would be very interesting to include*

See major comment regarding ecCKD. This paper concerns more specifically RRTMGP-NN, its refinement and online evaluation in the IFS. We emulate the smallest RRTMGP k-distributions available.

*- Line 121: I miss a benchmark on the OpenACC code in the paper*

Add a reference to PhD thesis (Ukkonen 2020) which includes CPU and GPU timings of RTE+RRTMGP-NN and a fairly recent version of RTE+RRTMGP.

*- Line 127: would the reverse be possible, to deliver full 3d fields per gpoint (see points above)?*

See major comment

*- Line 142: change "one person" in "the first author of this paper" so it is clear that you have it under control rather than that the reader wrongly conclude that there exists a dependency on an unknown person*

Changed as suggested, thank you

*- Line 207: what are square-root transformations? Please define this.*

Changed to: **"Firstly, the distributions of all outputs and some inputs were made more uniform by taking their N-th square root (a weaker form of log-scaling, see Table 1, Ukkonen et al., 2020)"**

*- Line 245-246: I do not understand this sentence, it is too abstract, please clarify this.*

Changed to **"In addition, the perturbation experiments must be designed and arranged in memory so that neighboring indices relate to the goal, which was minimizing the TOA and surface forcing errors of individual gases (for instance, a pair of adjacent experiments could be pre-industrial and present-day N2O)".**

*- Line 255, this sentence is ambiguous, are there 64 neurons spread over two layers, or two layers of 64 neurons?*

Changed to **"Two layers with 64 neurons in each layer"**

*- Line 285-286: it is 2023 now, please update the status of this.*

Good catch, changed to **"to which EMCWF's operational forecast migrated in October 2022".**

*- Line 293: this is a very nice result!*

Thanks!

*- Line 308: I think it is fair to mention that a better performance with respect to line-by-line is pure coincidence as you are not training against it*

The fact that the improvement over RRTMGP was quite robust (between training experiments) and, in the longwave upwelling fluxes, substantial, suggests that this is actually not a coincidence but because we use early stopping with respect to LBL. This is clarified in the next paragraph: "The result is therefore probably not a fluke, but attributable to using early-stopping based on a line-by-line benchmark".

For further clarification we added **"The exact mechanism for this is unclear, but it's plausible that the NN fits to physically correct signals in the data before it fits to reproduce RRTMGP errors with respect to LBL (or, the result is indeed a fluke; heating rate errors are slightly worse than for RRTMGP)"**

*- Figure 8, I don't think it is necessary to plot th reference lines in figure b and c, since they are zero.*

This is a matter of taste but we prefer to leave the reference lines on the 0-axis, and describe the reference model in the legend, to make it easier to differentiate negative and positive biases and clarify what the reference is (as it differs from b to c).

*- Line 392: "large share" is subjective, please share the numbers of your experiments. How expensive was the radiation in the running IFS and what did you gain?*

Added **"For instance, in coarse-resolution simulations using the ECHAM climate model, it accounted for half of the runtime of the atmospheric model (Cotronei et al. 2020). (In the IFS, radiation is currently only a few percent of the model runtime but this is largely due to it being called on a coarser grid and only every hour.)**

References:

Cotronei, A. and Slawig, T.: Single-precision arithmetic in ECHAM radiation reduces runtime and energy consumption, Geoscientific Model
Development, 13, 2783–2804, 2020

Hogan, R. J. and Matricardi, M.: A Tool for Generating Fast k-Distribution Gas-Optics Models for Weather and Climate Applications, Journal of Advances in Modeling Earth Systems, 14, e2022MS003 033, https://doi.org/https://doi.org/10.1029/2022MS003033, 2022

**REVIEWER 2**

*The manuscript presents a targeted approach to speeding up emulation of the radiation parameterization by focusing on the gas optic component of the RRTMG scheme. Overall, the manuscript is logically organized, easy to follow, and uses a smart strategy to limit the impact of including an emulator on the overall model. Additionally, I greatly appreciate that the authors included experiment results that embed the ML model during a prognostic simulation to evaluate the performance. While there are some minor suggestions to address, the manuscript is generally deserving of publication.*

*In the process of constructing emulators, it's possible to identify an effective model via physically-reasoned choices, trial-and-error, or a mix of both. I sympathize with the authors' point that the best model in this case involved some ad hoc decisions, but what matters for readers is to get insight into the most influential choices to facilitate their own implementation of the method. Where possible, the authors should elaborate on specific choices that significantly impacted the model performance with either sensitivity tests or explanations of the expected benefits. Specific examples are provided below along with other minor comments.*

Thank you for your comments which we feel are fair. Our previous paper - Ukkonen et al. 2020 (hereafter UK20) - was quite ad-hoc and unsystematic in its data generation especially, and this current paper builds on the previous one, so it makes sense that the reviewer picked up on this. The reason UK20 ended up with using multiple datasets that were manually supplemented in various ways was partially due to choice (aiming to provide NN gas optics models that could be used in both NWP and climate modeling contexts required diversity of data and greenhouse gas concentration scenarios, and sampling extremes) but partially unplanned: evaluation of radiative forcings with respect to individual gases revealed stubborn errors, and the data grew over time as we tried to improve the results. In this paper we have aimed to improve upon the iterative & ad-hoc aspect of UK20 by 1) incorporating the radiative forcing errors in the loss function and 2) monitoring the final metrics that we care about (heating rate, broadband flux and radiative forcing errors) during the training process and early stopping with respect to these metrics. Likewise, (3) the main dataset in this paper (a new CAMS dataset) is also constructed in a more systematic way, as described in section 3.1 and Fig. 2., partially by using methods from Hogan and Matricardi 2020, which we slightly elaborate in the revised paper.

Other elements were carried directly from UK20 - for instance data preprocessing and NN hyperparameters – and we refer the reader to UK20 for more details and justification of these. Regarding learning rate, I recall testing a few different values in UK20 but finding that the default Adam learning rate worked well. From a machine learning perspective, the gas optics problem is a relatively easy emulation problem as (unlike in radiative transfer) there is no non-locality or potential issues with energy conservation, and simple dense networks can easily predict optical properties with good accuracy. As such we feel justified in using manual tuning and carrying over hyperparameters and data processing methods from UK20, where they were found to work well. That being said, we aim to provide more clarity in the revised paper.

**Minor/specific comments**

- *Lines 173-181: I am not familiar with the preparation of the training data in the previous study but it seems some fairly complicated additions were made with the CAMS reanalysis data and selection of profiles. How much of a difference did the inclusion of the new data make in training performance or online performance?*
- *Line 183: "varied similarly to Hogan … " does not give much to go on for the data generation. Maybe supply a name of the variation method or what the target of the variation product is.*

The aim of the revised CAMS dataset was to more systematically sample the input distributions, especially its corners, in present-day climate for NWP applications, combined with systematically varying greenhouse gases following Hogan and Matricardi (2020). To clarify the latter we added:

"..varied similarly to Hogan and Matricardi (2020), **where the concentration of gases were individually varied across a large range, keeping other gases constant, except for CO2 , CH4 and N2O which were also perturbed in pairs to account for the overlap in their absorption spectra"**.

We do not have any numbers for how the inclusion of the CAMS data changes the performance (UK20 already had very diverse data) but its the main dataset we rely on, and thought it would be useful to improve upon it from UK20 since we no longer use RFMIP data for training.

- *Section 3.1: Perhaps a table listing the all data (in the appendix) would be helpful. List exactly how much and what data was used for what purpose.  I found the text here kind of confusing with the amount of different sources described.*

Added a table in the main text (Table 1).

- *Table 1: Is future - preindustrial RMS only being used for shortwave related to the error saturation discussed in Line 249? Might be worth spelling out in the text exactly which combinations of RMS are used for the loss and why longwave vs. shortwave have different selections of IRA components.*

I honestly don't remember the thinking behind that specific choice (this paper was originally submitted almost a year ago to another journal), but it was probably what you suggested: Future-PreIndustrial TOA flux could dominate the other metrics in the longwave. The selection of different metrics in LW and SW was ad-hoc but basically to emphasize heating rate errors in the shortwave, and radiative forcing errors in the longwave where the greenhouse effect is important – we now mention this in the revised paper.

- *Line 241: "after some testing" - say what the testing was and how the decision was made*

Changed to "**after testing a few different values and finding that these seemed to give a reasonable trade-off between heating rate errors and radiative forcing errors."**

- *Figure 3: I notice that the heating rate errors stay above 1 (even though the goal is at or below 1) when including the indirect optimization on forcing errors through the hybrid loss. Where any prognostic experiments run using the non-hybrid loss model? If so, how did it perform? Offhand, I'm not sure what larger N2O forcing errors would mean for an annual-scale simulation. Which errors are most important for an unbiased long simulation?*

Our goal was errors close to 1: "emulation errors should be smaller than parameterization error" means that RRTMGP – NN differences are smaller than RRTMGP – LBL differences. Good catch regarding the larger heating rate RMSE when using a hybrid loss. However in the CKDMIP evaluation of the longwave model (Appendix A1) the performance was acceptable: in present-day scenario the NN had an RMSE of 0.504 K d-1 while for RRTMGP it was 0.449, and for the future scenario the differences were smaller. Looking at Fig 3., the hybrid loss gives a huge improvement in forcing errors while the heating rate errors are relatively small, so this is a good trade-off. We did not perform any prognostic experiments with non-hybrid-loss models. Regarding the impact of N2O forcing errors, that is an interesting question that is beyond the scope of this paper to address, but certainly for shorter integrations the impact of even large relative errors should be small (In Fig 7. the maximum range of N2O forcing is within 1 W/m2) while for longer coupled simulations, where the energy imbalance accumulates over time, it's important that these errors are kept small.

- *Line 252: Was the validation loss of the primary predictand also tracked during training? If so, what was the behavior of the validation loss? I would expect that further refinement of the gas optics emulator would improve heating rate and forcing errors, so why do we need early stopping here? Signs of overfitting? The noisy loss curves have me wondering whether the learning rate might not be optimal near the end of training. This also ties into the training data selection and whether choices made in creation of the data truly provide the right data (i.e., spans necessary variability) to train the models.*

Note that the loss is not noisy: the loss is given by the red curves which are smooth. The hybrid loss includes the "expdiff" component (measuring differences in optical properties, i.e. the predictand, between different experiments) which is shown in the dashed red line, and this one is also smooth. We cannot directly optimize for the forcing errors (which dominate the variability in the solid blue line), at least not in an easy way, so we only monitor it and use it for early-stopping – this is why it's noisy, it's not actually used in training and has a non-linear relationship with optical properties. One factor may be that RMSE in optical properties will weight larger optical depths more, but errors at large optical depths may not matter much for fluxes since all radiation is likely to be absorbed in this case anyway. This may be a good reason to revise the loss function further in future work, but ideally one could train to directly optimize fluxes while predicting optical properties as discussed in the paper.

- *Line 254: Was any hyperparameter search done here or just in the previous paper? Would be nice to have confidence that the basics of capacity, learning rate, and batching are good. Or*

> *maybe add an explanation of why increased capacity might be necessary for this emulator problem vs. previous papers.*

See major comments regarding hyperparameters. Regarding the increase in number of neurons, while true, we failed to realize that the complexity is actually similar: the number of spectral terms was roughly halved, and in the LW we use one model instead of two – the number of total NN parameters is similar with UK20. Added "**(but the number of model parameters is similar; this is explained by the number of g-points being roughly half of the previous k-distributions)**

- *Line 264-272: This makes me think all the timings in Figure 4 use this enhanced version of the code.Is the reason for description of what's coming up in a future paper here because some reader's would be expecting different timings out of the old scheme? I note because this section confused me about whether speedups also included aspects of the code optimization vs. only the effects of the NN. It would be easier just to consider the effects of the NN.*
- *Figure 4: What's the point of showing the time of the other ecRad components? This was another reason I thought I was looking for improvements from code optimizations too. It's really about two numbers, the speed up in gas optics and the total speedup. Maybe not necessary to include a bar chart.*

While this work only reduces the runtime of gas optics, the more important metric is the acceleration of the radiation scheme overall. This depends on the cost of the other components, which we have reduced in an another paper that we have just submitted to JAMES – the timings using the optimized code are more relevant as the optimizations will feature in an official ecRad version in the future. Regarding the figure we believe it's useful to see a cost of all the components to get a sense of their relative importance and facilitate comparisons with other papers and other codes.

- *Was there any attempt at measuring/estimating how much of a speedup using the NN would convey when GPU was available? Could be interesting as a standalone offline comparison if it's not too hard.*

We added a reference to section in a PhD thesis which shows GPU and CPU timings of RTE+RRTMGP and its NN version:

"**(While efficient, the speed advantage over version 1.5 of RTE+RRTMGP is actually much smaller on GPU than CPU, as shown in section 3.2 of Ukkonen, 2022b). "**

The timings shown in the thesis use the reduced k-distributions. The small speedup on GPUs is perhaps surprising but it's probably because on CPU, reference RRTMGP really suffers from a lack of vectorization, whereas this is not an issue on GPU. I understand that the GPU code of RTE+RRTMGP is well optimized (in collaboration with NVIDIA) and its tough to beat it even with NNs (which are efficient but perform more floating point operations).

References:

Ukkonen, P.: Improving the trade-off between accuracy and efficiency of atmospheric radiative transfer computations by using machine

learning and code optimization, Ph.D. thesis, School of The Faculty of Science, University of Copenhagen, 2022b.